# Systemic Therapy Development in Von Hippel–Lindau Disease: An Outsized Contribution from an Orphan Disease

**DOI:** 10.3390/cancers14215313

**Published:** 2022-10-28

**Authors:** Vivek Narayan, Eric Jonasch

**Affiliations:** 1Division of Hematology/Medical Oncology, University of Pennsylvania, Abramson Cancer Center, Philadelphia, PA 19104, USA; 2Genitourinary Medical Oncology, MD Anderson Cancer Center, Houston, TX 77030, USA

**Keywords:** von Hippel–Lindau, HIF2 inhibitor, renal cell carcinoma, hemangioblastoma, systemic therapy

## Abstract

**Simple Summary:**

This review highlights the development of systemic anti-neoplastic therapies for the treatment of patients with von Hippel–Lindau disease, culminating with the recent approval of the first systemic agent for this disease.

**Abstract:**

Over the last several decades, an improved understanding of von Hippel–Lindau disease and its underlying biology has informed the successful development of numerous anti-cancer agents, particularly for the treatment of advanced renal cell carcinoma. Most recently, this has culminated in the first regulatory approval for a systemic therapy for VHL disease-associated neoplasms. This review will trace the clinical development of systemic therapies for VHL disease and additionally highlight anticipated challenges and opportunities for future VHL systemic therapy.

## 1. Introduction

Over a century ago, the identification of a familial pattern of retinal hemangioblastomas by ophthalmologist Eugen von Hippel, and the subsequent association of this disorder with an increased risk of both benign and malignant systemic neoplasms by neuropathologist Arvid Lindau, led to the discovery of von Hippel–Lindau (VHL) disease [1]. Officially coined in the 1930s, VHL disease represents a hereditary, autosomal-dominant condition that may manifest with a variety of tumor types including central nervous system (CNS) and retinal hemangioblastomas, pancreatic neuroendocrine tumors (pNETs), pheochromocytomas, inner ear endolymphatic tumors, epididymal or broad ligament cystadenomas, clear cell renal cell carcinomas (RCC), and/or pancreatic or renal cystic lesions [2]. Occurring in approximately one in 35,000 live births, VHL disease is caused by a germline mutation and/or deletion in the *VHL* tumor suppressor gene, located on the short arm of chromosome 3 [3,4]. VHL disease can be diagnosed based on clinical criteria incorporating the presence of syndromic tumors, with or without a familial VHL disease history [5,6], and phenotypic subtypes have been assigned depending on the absence or presence of pheochromocytoma (Type 1 or Type 2 disease, respectively) [7,8].

Historically, the management of VHL disease-associated neoplasms has entailed careful imaging and clinical surveillance, with the deployment of serial local interventions including surgical, ablative, or ophthalmologic procedures in the event of significant or symptomatic growth. In particular, RCCs and CNS hemangioblastomas often require serial high-risk, invasive interventions, thereby contributing to significant physical and psychologic morbidity for patients. Patients with VHL disease have a lifetime risk of RCC malignancies of up to 70% and face the potential for metastatic dissemination of RCC if tumors grow beyond 3 cm in diameter [9,10,11]. Although surgical resection can be curative therapy for VHL-associated RCC, the risk of local recurrences and/or incident tumors in the remaining or contralateral kidney is inherently elevated, thereby limiting the potential durability of local interventions [12]. Thus, the management of VHL disease-associated RCC has entailed the use of active surveillance for RCC tumors <3 cm in size, with selective nephron-sparing interventions such as partial nephrectomy or percutaneous ablative procedures, reserved for the treatment of tumors beyond the 3 cm threshold or for rapidly growing lesions [8,13]. Nevertheless, the recurrent nature of RCC in VHL disease can lead to significant cumulative patient morbidity including potential metastatic dissemination or the need for hemodialysis or renal transplantation [14]. Similarly, both CNS (brain and spinal cord) and retinal hemangioblastomas demonstrate recurrence and progression, resulting in serial interventions and significant cumulative morbidity [15].

Given the requirement for serial invasive procedures to manage disease-associated neoplasms throughout a VHL patient’s lifetime, effective systemic therapy for VHL disease could impart significant clinical benefit by reducing the need for invasive interventions and the resulting iatrogenic morbidity. In addition, the unifying genetic predisposition and biology of VHL disease-associated neoplasms lends well to potential therapeutic targeting [7]. Indeed, an improved understanding of the *VHL* gene and its downstream gene products has informed the successful development of numerous anti-cancer agents for sporadic clear cell RCC and ultimately culminated in regulatory approval by the United States Food and Drug Administration (US FDA) of the first systemic agent for patients with VHL disease-associated RCC, pNETs, or CNS hemangioblastomas [16]. This review will trace the clinical development of systemic therapies for VHL disease and additionally highlight anticipated challenges and opportunities for future VHL systemic therapy.

## 2. VHL Biology and Inappropriate Angiogenesis

The *VHL* gene was identified in 1993, and the structure and function of the *VHL* gene product (pVHL) were elucidated over the subsequent decade [3]. pVHL forms a ternary complex with the transcription elongation factors elongin C and B, forming the VCB complex [17,18]. Through cross-stabilization between pVHL and elongins B and C, the VCB complex is resistant to proteasomal degradation [19,20,21]. pVHL plays a key role in the cellular signaling resulting from changes in oxygen tension by functioning as the substrate recognition subunit of an E3 ubiquitin ligase complex that targets the α subunit of the heterodimeric hypoxia-inducible factor (HIF) transcription factor for proteasome degradation [22]. Under normoxic conditions, pVHL recognizes the oxygen-dependent prolyl-hydroxylation of HIFα and targets it for ubiquitylation [7,23]. Polyubiquitylated HIFs are subsequently sequestered and degraded by the cellular proteosome. However, under hypoxic conditions, or in the absence of functional pVHL due to a VHL-disease associated mutation, HIFα constitutively accumulates and forms heterodimers with HIF1β [24]. These heterodimers subsequently translocate to the nucleus and bind to hypoxia-response elements (HREs), inducing downstream gene expression and promoting cellular adaptation to acute or chronic hypoxia [24,25].

The HIF transcriptional complex promotes the expression of over 100 proteins, and key factors regulated by the pVHL–HIF pathway include HRE-related pro-angiogenic proteins as well as proteins involved in cellular growth and metabolism. In particular, HIF-mediated transcription induces the gene expression of the vascular endothelial growth factor (VEGF), platelet-derived growth factor (PDGF), cyclin D1, and glucose transporter 1 (GLUT1) [26,27]. Due to the downstream inappropriate pro-angiogenic upregulation occurring in the setting of dysfunctional pVHL, the highly vascular neoplasms present in VHL disease are known to overproduce such angiogenic peptides [28,29]. Moreover, consistent with the tumorigenesis implicated in hereditary VHL disease, somatic gene aberrations are identified in the vast majority (up to 90%) of sporadic clear cell RCCs [30,31,32]. Indeed, this understanding of VHL biology and the inappropriate pro-angiogenic signaling resulting from VHL inactivation in clear cell RCC has revolutionized the systemic therapy of advanced RCC over the past two decades with the clinical development and eventual regulatory approvals of numerous VEGF receptor tyrosine kinase inhibitors (VEGFR TKIs) including sunitinib, pazopanib, and axitinib.

## 3. Anti-Angiogenic Systemic Therapies in VHL Disease

Given the shared pVHL–HIF pathway tumor biology implicated in both VHL-associated disease and sporadic RCC, its functional consequence of inappropriate angiogenesis, and the proven anti-tumor efficacy and regulatory approvals of VEGFR TKI therapies in sporadic advanced clear cell RCC, these antiangiogenic agents were among the first tested for the systemic treatment of patients with VHL disease-associated RCC and other neoplasms (Table 1). In a prospective, open-label, single-arm phase 2 clinical trial, 15 patients with genetically-confirmed VHL disease and at least one measurable VHL disease-associated lesion (CNS hemangioblastoma, RCC, pancreatic cysts or NETs) received sunitinib 50 mg daily on a 28-day on/14-day off schedule for up to four treatment cycles (total duration of 6 months) [33]. The primary objective was to evaluate the safety of sunitinib, and the secondary objectives included efficacy evaluation assessed by radiographic response. Patients may also have had retinal hemangioblastomas, which were followed by regular direct ophthalmoscopy. While all patients received at least two treatment cycles (50% of planned therapy), six patients (40%) discontinued therapy prior to the completion of four treatment cycles. Grade 3 toxicity occurred in five patients (33%) and included fatigue (33%), hand-foot syndrome (13%), and nausea (13%). Sunitinib dose reduction was required in 10 patients (66%) (37.5 mg daily in six patients; 25 mg daily in four patients). The observed anti-tumor response appeared to be lesion-dependent, with six out of 18 (33%) evaluable RCCs demonstrating a partial response (PR), and no responses observed in 21 evaluable CNS hemangioblastomas. A best response of stable disease (SD) occurred in five evaluable pNETs, and none of the seven retinal lesions demonstrated response per ophthalmoscopy [33]. Similarly, in a single arm prospective study of five patients with genetically-confirmed VHL disease treated with the same dose and schedule of sunitinib, the best observed response was SD and treatment was limited by unacceptable toxicity (occurring in three out of five patients within 6 months of treatment initiation) [34]. Overall, these findings indicate that while sunitinib anti-angiogenic therapy offered some anti-tumor response in VHL disease-associated RCC, the minimal efficacy against extra-renal disease manifestations and the significant treatment-related toxicity limited its effective long-term use in this setting.

The analysis of sporadic RCC and VHL-associated hemangioblastoma archival tissues was subsequently conducted in an effort to evaluate the differential response observed between RCC and CNS hemangioblastoma lesions exposed to sunitinib in VHL disease [33]. Notably, while RCC tissues demonstrated higher levels of phosphorylated VEGFR-2 when compared to hemangioblastomas, the protein levels of fibroblast growth factor receptor 3 (FGFR3) and FGFR ligand (FGFR substrate 2) were higher in hemangioblastomas relative to RCC [33]. This intriguing finding provided a mechanistic rationale for the clinical testing of dovitinib, a multi-kinase inhibitor targeting FGFR, VEGFR, and PDGFR, in patients with genetically-confirmed VHL and at least one measurable hemangioblastoma [35]. Unfortunately, the study was terminated following the treatment of only six patients due to intolerable toxicity. Three of six patients (50%) discontinued therapy due to adverse effects including maculopapular rash, vomiting, and dyspnea, despite dose reduction and supportive measures. The best response observed was SD in hemangioblastomas, which occurred in five out of six patients [35].

The largest prospective study to date evaluating a VEGFR-directed approach for patients with VHL disease was a non-randomized phase 2 single-center open label clinical trial of pazopanib [36]. Thirty-one eligible patients with genetically-confirmed or clinically-defined VHL disease received pazopanib of 800 mg daily for a planned treatment duration of at least 24 weeks. Co-primary endpoints of the study were safety and objective response rate. Common adverse events included fatigue, hypertension, and diarrhea, and four patients (13%) required treatment discontinuation due to grade ≥3 transaminitis. An additional three patients (10%) discontinued due to the intolerance of lower-grade cumulative toxicities. Ultimately, 21 patients (67%) required pazopanib dose reduction due to toxicity. However, pazopanib therapy demonstrated clear therapeutic activity in VHL disease. Across the 59 total RCC tumors, a PR was observed in 29 (49%) and a complete response (CR) occurred in two tumors (3%). While pazopanib achieved a PR in nine out of 17 (53%) pancreatic tumors, only two out of 49 (4%) CNS hemangioblastomas demonstrated PR. Only seven patients (23%) elected to remain on pazopanib therapy beyond 24 weeks. Therefore, while pazopanib demonstrated clinical activity in VHL disease, particularly in RCC tumors, the clinical benefit for patients was limited by treatment-related toxicity and suboptimal efficacy toward VHL-disease associated hemangioblastoma.

## 4. Development of HIF2α Inhibition as Cancer Therapy

As previously described, pVHL exerts its tumor suppressor functions via HIFα regulation. Indeed, pathogenic VHL gene mutations have largely been identified to impair the interaction between pVHL and HIF [37,38,39]. HIF transcription factors are heterodimers consisting of an α subunit (HIF1α, HIF2α, HIF3α), and a β subunit (HIF1β/aryl hydrocarbon receptor nuclease translocator (ARNT)). Among the various HIFα subunits, HIF2α is more commonly expressed in renal, endothelial, and lung cells, and is most implicated in clear cell RCC oncogenesis [40,41]. Upon dimerizing with ARNT, HIF2α binds to hypoxic response elements, promoting angiogenesis and cellular growth. Therefore, specific targeting of HIF2α-related transcription activity is hypothesized as a potentially effective therapeutic strategy for VHL disease-associated neoplasms, particularly in clear cell RCC.

As HIF2 and ARNT heterodimer formation and nuclear translocation induces pro-angiogenic gene transcription and cellular growth, novel drug development targeting the HIF2/ARNT interaction was actively pursued. However, as a transcription factor, HIF2α therapeutic targeting poses several challenges [42]. Fortunately, the breakthrough discovery of a therapeutic vulnerability in the HIF2α Per-ARNT-Sim (PAS) B domain, which enables stable HIF2a/ARNT heterodimer formation, has opened the door for targeted drug development [43]. Using a structure-based drug design process targeting the HIF2α PAS B binding domain, thereby inducing a conformational change and limiting HIF2α/ARNT interaction, the initial candidate HIF2 α inhibitor compounds, PT2385 and PT2399, were developed [44,45,46]. In a phase 1 dose-escalation clinical trial, PT2385 was evaluated in 26 patients with sporadic advanced RCC [47]. In this treatment-refractory patient population (median number of prior systemic therapy lines was 4, range 1–7), dose-limiting toxicity was not observed, and a maximum tolerated dose was not identified. Drug exposure did not significantly increase beyond 800 mg twice per day. Rapid reductions in plasma erythropoietin (EPO), which is secreted by renal interstitial fibroblasts and is regulated by HIF2α, served as a pharmacodynamic measure of the successful attenuation of HIF2α target HRE gene expression and was observed at all dose levels. However, dose levels evaluated above 800 mg twice a day did not result in greater EPO reductions. Therefore, based on the overall drug exposure and pharmacodynamic data, 800 mg twice daily was selected as the recommended phase 2 dose (RP2D).

In a subsequent Phase 1 dose-expansion, 25 patients with advanced RCC were treated at the PT2385 RP2D [47]. Among the 51 total RCC patients treated across dose-escalation and expansion, the treatment-related toxicities were mostly low grade. Notable treatment-related adverse events included anemia (grade 1/2 35%; grade 3 10%), fatigue (grade 1/2 37%), and hypoxia (grade 3 10%, including two patients with a serious AE at the 1200 mg twice daily dose-level). The observed objective response rate with PT2385 was 14% (2% CR, 12% PR). Twenty-six patients (52%) had SD including 42% with SD ≥4 months. *VHL* was mutated in tumors from five out of nine patients with available tissue samples, and *VHL* mutations were enriched among patients with SD [48]. In the gene set enrichment analysis by RNA-Seq in thee available paired baseline and on-treatment tumor samples, PT2385 appeared to significantly decrease the HIF2α target gene expression levels [45,48]. In the pharmacokinetic analysis, significant inter-patient variability in drug exposure was identified, with higher drug exposure associated with improved RCC progression-free survival [47]. This pharmacokinetic variability was attributed to hepatic metabolism via UDP-glucuronosyltransferase UGT2B17 glucuronidation, which is known to have inter-patient differential expression.

This improved understanding of PT2385 metabolism and pharmacokinetics and its correlation with anti-tumor efficacy, subsequently led to the development of the next-generation HIF2α inhibitor PT2977 (MK-6482, subsequently named belzutifan), which displayed decreased glucuronidation, an improved pharmacokinetic profile, and enhanced potency relative to its parent compound [49]. In a phase 1 dose-escalation clinical trial evaluating belzutifan in 43 patients with advanced solid tumors, no dose-limiting toxicity was observed at dose levels up to 160 once daily of belzutifan [50]. However, treatment-related dose-limiting toxicities (grade 4 thrombocytopenia, grade 3 hypoxia) were observed at higher dose levels. Exposure to belzutifan increased with doses up to 120 mg once daily. Similar to PT2385, reductions in plasma EPO were observed at all dose levels, and the plasma EPO concentrations correlated with the drug exposure levels. However, dose levels above 120 mg once daily did not correlate with further decreases in the EPO concentration. Therefore, a daily dose of 120 mg PO daily was selected as the belzutifan RP2D [50]. Among the 55 patients with clear cell RCC treated at the RP2D level in a dose-expansion cohort, common all-grade AEs included anemia (76%), fatigue (71%), and dyspnea (49%). Fifteen patients (27%) developed grade 3 anemia, typically managed with exogenous EPO administration and/or blood transfusion rather than dose reduction or drug discontinuation. Nine patients (16%) developed grade 3 hypoxia including two patients requiring discontinuation of therapy. The objective response rate was 25% (all PRs) in the cohort of 55 clear cell RCC patients treated with belzutifan 120 mg daily dosing. Median progression-free survival was 14.5 months, and 19 patients (35%) remained on belzutifan for longer than 12 months [50].

## 5. HIF2α Inhibition in VHL Disease

Since an understanding of VHL disease biology and pathogenesis led to the preclinical development of drugs targeting HIF2α transcription activity, VHL disease represents a model disease system for the clinical testing of belzutifan. The clinical activity of belzutifan in VHL disease was specifically tested in a phase 2, non-randomized, open-label clinical trial of 61 patients with germline VHL alteration and at least one measurable, non-metastatic primary RCC [51]. Eligible patients did not have a RCC lesion larger than 3 cm that necessitated immediate surgical resection to mitigate metastatic risk. Belzutifan was administered at a starting dose of 120 mg daily, and the primary study endpoint was the objective response rate of RCC lesions. Secondary endpoints included safety and the objective response rate in non-RCC VHL disease-associated neoplasms (including CNS hemangioblastomas and pancreatic lesions). Consistent with the serial invasive interventions typically required for the management of VHL disease, 40 patients (66%) had undergone prior partial or radical nephrectomy, and 47 patients (77%) had undergone prior neurosurgical intervention. After a median follow-up period of 21.8 months (range 20.2–30.1), a total of 56 patients (92%) had an observed reduction in the sum of RCC target lesions including 30 patients (49%) with a confirmed PR. An additional 30 patients (49%) had SD. At the time of the initial data cut-off, the median duration of response had not been not reached (2.8+–22.3+ months), and 54 patients (89%) remained on belzutifan therapy. Similar anti-tumor efficacy was observed in non-RCC VHL disease-associated neoplasms, with 47 of 61 patients (77%) having a confirmed response in pancreatic lesions (including a 91% confirmed response rate in 20 patients with pancreatic NETs). Among the 50 patients with CNS hemangioblastomas, the objective response rate was 30% including three patients (6%) with complete radiographic responses. Treatment-related toxicity was mostly low grade (grade 1 or 2), and belzutifan dose-reduction was required in only nine patients (15%). Treatment-related grade 3 events occurred in nine patients (15%) and included anemia (8%), fatigue (5%), and hypertension (8%). Although four patients (7%) received blood transfusion and 12 patients (20%) received erythropoietin-stimulating agents, treatment-related anemia typically stabilized without intervention in most patients. No treatment-related grade 4 or 5 events occurred.

Overall, these clinical findings compared favorably to the observed experience with VEGFR TKI therapies for the treatment of VHL disease. For example, while pazopanib TKI therapy demonstrated comparable objective response rates (42%) in VHL disease-associated neoplasms (RCC, pancreatic, and CNS neoplasms), the treatment exposure was limited by more significant toxicity (23% toxicity-related discontinuation rate and 68% dose-reduction rate) [36]. The patient acceptability and ongoing clinical efficacy of belzutifan was reinforced with longer-term follow-up data of the phase 2 trial in VHL disease [52]. After a minimum follow-up period of 24 months (range 27.6–37.5), 50 out of 61 patients (82%) remained on belzutifan therapy. Among patients discontinuing study therapy (N = 11), treatment discontinuation was due to the progression of RCC neoplasms (N = 4), patient decision (N = 4), adverse events (N = 2), or unrelated patient death (N = 1). The RCC objective response rate improved to 59% including 3% CR (Table 1). The median duration of response was still not reached, with some patients having an objective response lasting greater than two years. In non-RCC VHL disease-associated lesions, the objective response in pancreatic NETs (N = 20) and CNS hemangioblastomas (N = 50) remained 90% and increased to 38%, respectively. Similarly, among the 12 patients (16 total eyes) with baseline retinal hemangioblastomas, all demonstrated ophthalmologic improvement. No new safety signals emerged with this longer-term follow-up. Ultimately, these overall findings led to the regulatory approval of belzutifan in August 2021 as the first systemic treatment for adult patients with VHL disease who require therapy for RCC, CNS hemangioblastomas, or pancreatic NETs, not requiring immediate surgery [16].

## 6. Future Directions for Systemic Therapy in VHL Disease

### 6.1. Remaining Clinical Questions

The approval by the U.S. FDA of the first oral HIF2α inhibitor as systemic therapy for patients with VHL represents a landmark event in the management of this disease. The availability of an effective and tolerable systemic therapy will undoubtedly influence the patient and provider decisions regarding the management of VHL disease-associated neoplasms, especially given the significant cumulative morbidity associated with serial high-risk invasive surgical or ablative interventions. However, the availability and clinical use of belzutifan also raises several key clinical questions regarding the optimal use of this agent. First, an improved understanding of the natural history of VHL-disease associated neoplasms including variation in interpatient and organ-specific growth kinetics could assist with clinical decision making regarding the timing of belzutifan initiation. Furthermore, how the use of belzutifan affects organ-specific growth rates (ex: RCC versus CNS hemangioblastoma) over longer follow-up periods, and to what extent the discontinuation of belzutifan may affect the organ-specific lesion growth rates, are unknown. These findings could have clear practical implications for belzutifan administration including the potential utility of intermittent, long-term treatment strategies. Second, correlations between VHL disease genotype and disease manifestations (phenotype) and organ-specific responses to belzutifan therapy should be elucidated. Third, although belzutifan has demonstrated generally acceptable toxicity thus far, longer safety and tolerability follow-up will be critical to evaluate the long-term risks and benefits for individual patients. Given the recent U.S. regulatory approval of belzutifan for VHL patients and the increasing population of non-trial patients who will be receiving this agent, there is now a unique opportunity to gather larger, real-world datasets regarding the efficacy, safety, patient reported outcome measures, and genotype–phenotype response correlations with belzutifan. Indeed, a multi-center initiative across Clinical Care Centers recognized by the VHL Alliance is planned to more efficiently gather retrospective and prospective longitudinal data on patients with VHL. Such datasets will offer improved insight into the natural history of the disease, potential genotype–phenotype correlations, and the impact of belzutifan initiation. Moreover, as the available clinical trial datasets on systemic therapy in VHL disease do not provide longitudinal assessment of health-related quality of life measures, these prospective real-world, multi-center datasets afford a key opportunity for assessing these important patient-centered measures. Fourth, as VHL disease is attributed to underlying pathogenic hereditary gene alterations targetable by belzutifan therapy, the application of a tumor ‘interception’ strategy with therapeutic HIF2α inhibition in patients with known genetic predisposition, but without prevalent VHL disease-associated manifestations, is highly attractive [53,54]. However, an improved long-term safety and efficacy dataset, particularly in younger patient populations, would be important to assess the potential risks and benefits of such an approach. Finally, as belzutifan expands to non-clinical trial populations including potential international health systems, issues related to access, cost barriers, and long-term financial toxicity may arise. Given the multiple potential management options available to some patients with progressing VHL disease-associated lesions, comparative cost-effectiveness studies with belzutifan may be warranted.

### 6.2. Mechanisms of Resistance

As belzutifan therapy is employed for longer treatment durations in patients with VHL disease, the potential mechanisms of resistance to HIF2α inhibition will warrant further investigation. In early phase studies of PT2385 for the treatment of patients with advanced clear cell RCC, whole exome sequencing (WES) of an oligoprogressive metastatic lesion indicated the presence of an acquired gatekeeper mutation in HIF2α, leading to therapeutic resistance [48]. WES identified a c.968G>A substitution in HIF2α that was not present in the baseline, pre-treatment tumor samples from responding metastatic sites. Notably, the c.968G>A substitution translates to a p.Gly323Glu (G323E) mutation, thereby directly altering the Gly323 residue located within the PT2385 drug binding pocket. In the functional assays and RNA-sequencing studies, the G323E substitution prevented HIF2 dissociation by PT2385 and interfered with PT2385-mediated downregulation of HIF2α target gene expression in the progressive metastasis, indicating that this represents an acquired gatekeeper resistance mutation in HIF2α [48]. As belzutifan binds to the same drug pocket as PT2385, similar acquired resistance mutations may be implicated. However, how such acquired mutations may contribute to treatment resistance in patients with VHL disease and germline pathogenic alterations remain to be seen.

pVHL may also serve several non-canonical functions beyond the regulation of HIF2α transcriptional signaling [55]. In addition, HIF2α inhibition can downregulate oncogenic genes independent of VEGF angiogenic signaling including *CCND1*. In preclinical analyses evaluating potential factors contributing to tumor growth in VHL-null Drosophila and human clear cell RCC, synthetic lethality was discovered between VHL and the cell cycle regulators cyclin-dependent kinases 4 and 6 (CDK4/6) [56]. Importantly, while HIF2α transcription induced the expression of cyclin D1, a CDK4/6 cell cycle partner, HIF2α activity was not required for the proliferative effects of CDK 4/6 in VHL-null cells. Therefore, given the potential benefit in both HIF2α-dependent and HIF2α-independent clear cell RCC, the use of CDK4/6 inhibitors was hypothesized to have a synergistic therapeutic benefit in combination with HIF2α inhibitors. A randomized phase 1/2 clinical trial of belzutifan with or without the CDK4/6 inhibitor palbociclib is currently planned for patients with advanced clear cell RCC to clinically evaluate this hypothesis (NCT05468697) (Table 2).

### 6.3. Application of HIF2α Inhibition beyond VHL Disease

Just as an improved understanding of VHL disease biology led to the successful development of anti-angiogenic VEGFR TKIs for the treatment of sporadic clear cell RCC and other advanced malignancies, the development of the HIF2α inhibitor belzutifan will likely inform novel treatment approaches for other non-VHL disease-related malignancies. Due to the shared pathogenesis between VHL disease and sporadic clear cell RCC as well as the aforementioned promising early efficacy signals, belzutifan is currently under late-stage clinical testing for the treatment of sporadic clear cell RCC including in both treatment-refractory advanced disease (NCT04195750) and post-nephrectomy adjuvant treatment settings (NCT05239728) (Table 2).

The exciting potential for targeted HIF2α inhibition outside of VHL disease was also highlighted in a recent report describing the use of belzutifan in an adolescent patient with Pacak–Zhuang syndrome, a tumor predisposition syndrome caused by a gain-of-function mutation in the HIF2α *EPAS* gene, most often via post-zygotic somatic mosaicism [53]. In patients with Pacak–Zhuang syndrome, the pathologic accumulation of HIF2α and subsequent transcriptional upregulation of HREs manifests as polycythemia with inappropriately elevated EPO, hypertension, and functional norepinephrine-secreting paragangliomas during early adolescence. In a case report of an adolescent patient with polycythemia and a high burden of functional paragangliomas, belzutifan led to a marked reduction in paraganglioma biochemical secretion (norepinephrine, chromogranin A) and radiographic regression of paraganglioma tumors, a cessation of anti-hypertensive agents, and a reduction in EPO and hemoglobin levels, obviating the need for therapeutic phlebotomy [53]. This activity raises the prospect for belzutifan activity in ‘pseudohypoxia-related’ paragangliomas including VHL-disease related paraganglioma and tricarboxylic acid cycle-related disease (such as SDHx and FH-related disease). A phase 2 non-randomized clinical trial of belzutifan in patients with advanced pheochromocytoma/paraganglioma including patients with VHL-related and SDHx-related disease is ongoing (NCT04924075). Notably, given the high objective response rates observed in VHL patients with pancreatic NETs, and the known angiogenic signaling implicated in sporadic NETs, a cohort of patients with sporadic advanced pancreatic NETs was also included.

Similarly, the observed benefit of belzutifan in VHL disease may not be restricted to patients harboring a pathogenic germline variant in *VHL*. A recent study evaluating a proband with clinically diagnosed VHL disease, but without a detectable *VHL* mutation, identified a de novo pathogenic variant within the gene encoding elongin C (ELOC). As elongin C plays a key functional role in the pVHL VCB complex, pathogenic ELOC variants were identified as a novel cause for VHL disease, the expression of hypoxia-response elements, and RCC pathogenesis. Thus, belzutifan therapy may offer clinical benefit in such novel genetic etiologies for VHL disease-associated phenotypic neoplasms.

## 7. Conclusions

As highlighted in this review, an improved understanding of VHL biology over the last several decades has paid large dividends for the treatment of both VHL disease-associated neoplasms and sporadic malignancies. Most recently, the clinical development of novel targeted HIF2α inhibitors is likely to continue this tradition. While representing the first approved systemic therapy for VHL disease-associated neoplasms, thereby significantly altering the management landscape for patients with VHL disease, belzutifan will likely additionally contribute to future systemic therapy approaches for non-VHL disease malignancies.

## Figures and Tables

**Table 1 cancers-14-05313-t001:** Prospective studies of systemic therapy for VHL disease-associated neoplasms.

Drug	# of Patients	Planned Treatment Duration	VHL Lesions	Best Radiographic Response				Grade ≥ 3 AE (%)	Reference
				CR	PR	SD	PD		
**Sunitinib**	15	6 months	RCC (N = 21)		33%	56%	11%	Fatigue 5% HFS 13% Nausea 13% Hypertension 7%	33
			CNS Hb (N = 18)			91%	9%		
			Pancreatic NET (N = 5)			100%			
			Retinal angioma (N = 7)	100% with stable findings					
**Sunitinib**	5	Indefinite	RCC (N = 5)			100%			34
**Dovitinib**	6	Indefinite	CNS Hb			100%		Rash (17%)	35
**Pazopanib**	31	6 months	RCC (N = 59)	3%	49%	47%		AST increase (10%) ALT increase (13%) Proteinuria (3%)	36
			CNS Hb (N = 49)		4%	96%			
			Pancreatic Lesions (N = 17)		53%	47%			
**Belzutifan**	61	Indefinite	RCC (N = 61)	3%	56%	39%		Anemia 10% Fatigue 5%	51, 52
			CNS Hb (N = 50)	6%	32%	52%	6%		
			Pancreatic Lesions (N = 61)	15%	66%	18%			
			Pancreatic NETs (N = 20)	15%	75%	10%			
			Retinal angioma (N = 16)	100% with improvement					

Abbreviations: VHL, von Hippel–Lindau; NET, neuroendocrine tumor; AE, adverse event; CNS, central nervous system; Hb, hemangioblastoma; RCC, renal cell carcinoma; CR, complete response; PR, partial response; SD, stable disease; PD, progressive disease.

**Table 2 cancers-14-05313-t002:** Ongoing and planned clinical trials of belzutifan.

Trial Name	Trial Phase	Planned Accrual (N)	Eligible Tumor Type(s)	Trial Identifier
Belzutifan for the treatment of advanced pheochromocytoma/paraganglioma (PPGL), pancreatic neuroendocrine tumor (pNET), or von Hippel–Lindau (VHL) disease-associated tumors (MK-6482-015)	2	232	Pheochromocytoma/Paraganglioma pNET VHL disease-associated tumors (at least 1 measurable PPGL or pNET)	NCT04924075
A study of belzutifan plus versus placebo plus pembrolizumab in participants with clear cell renal cell carcinoma post nephrectomy (MK-6482-022)	3	1600	Clear cell RCC (post-nephrectomy)	NCT05239728
A study of belzutifan in combination with palbociclib versus belzutifan monotherapy in participants with advanced renal cell carcinoma (MK-6482-024)	1/2	180	Advanced Clear Cell RCC	NCT05468697
A study of belzutifan versus everolimus in participants with advanced renal cell carcinoma (MK-6482-005)	3	736	Advanced Clear Cell RCC	NCT04195750
A study of pembrolizumab in combination with belzutifan and lenvatinib, or pembrolizumab/quavonlimab in combination with lenvatinib, versus pembrolizumab and lenvatinib, for treatment of advanced clear cell renal cell carcinoma (MK-6482-012)	2	1431	Advanced Clear Cell RCC	NCT04736706
Pembrolizumab plus lenvatinib in combination with belzutifan in solid tumors (MK-6482-016)	2	730	Hepatocellular carcinoma Colorectal cancer Pancreatic ductal adenocarcinoma Biliary tract cancer Endometrial cancer Esophageal squamous cell carcinoma	NCT04976634

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
