# Peer review of "Systemic Therapy Development in Von Hippel–Lindau Disease: An Outsized Contribution from an Orphan Disease"

_cancers, 2022, doi:10.3390/cancers14215313_

Round 1

Reviewer 1 Report

I think this is an excellent, well written, comprehensive review which is timely with respect to current clinical practice.

I have just 2 comments.

1) The authors do not mention the recent publication of Elongin C (ELOC/TCEB1)-associated von Hippel-Lindau disease (https://pubmed.ncbi.nlm.nih.gov/35323939/)

and I wondered whether a comment on this and the potential of therapeutic interventions also affecting this pathway would be relevant.

2) In terms of implementation of Belzutifan into healthcare systems there is no mention of the financial barriers that may present internationally which could impact on availability and a comment on the need to show cost effectiveness as well as clinical efficacy to future proof patient access might be helpful in the section "Future Directions for Systemic Therapy in VHL Disease"

Otherwise I thought this was a very comprehensive helpful review

Author Response

Please see that attachment.

Reviewer 2 Report

The authors provided a well written and thorough overview of the available studies and literature. 

Unfortunately, no data on the quality of life assessments during the individual studies are described. Please provide, especially on the recent belzutifan studies. If there is no data, this should also be described.

The authors state that belzutifan is the first approved drug for VHL. This is only by the FDA and not in Europe (EMA). This should be specified. 
